# Improving Neural Network Surface Processing with Principal Curvatures

**Josquin Harrison**
Inria
Sophia Antipolis
`josquin.harrison@inria.fr`

**James Benn**
Inria
Sophia Antipolis
`james.benn@inria.fr`

**Maxime Sermesant**
Inria
Sophia Antipolis
`maxime.sermesant@inria.fr`

## Abstract

The modern study and use of surfaces is a research topic grounded in centuries of mathematical and empirical inquiry. From a mathematical point of view, curvature is an invariant that characterises the intrinsic geometry and the extrinsic shape of a surface. Yet, in modern applications the focus has shifted away from finding expressive representations of surfaces, and towards the design of efficient neural network architectures to process them. The literature suggests a tendency to either overlook the representation of the processed surface, or use overcomplicated representations whose ability to capture the essential features of a surface is opaque. We propose using curvature as the input of neural network architectures for surface processing, and explore this proposition through experiments making use of the *shape operator*. Our results show that using curvature as input leads to significant a increase in performance on segmentation and classification tasks, while allowing far less computational overhead than current methods.

## 1 Introduction

Surfaces are a natural representation for many real world objects ranging from organs and organisms to archaeological artefacts. They are also a central tool in virtual environments such as computer games, or computer-aided design. This ubiquity has resulted in a large body of work dedicated to mathematical methods developed for the efficient use of surfaces, as well as their analysis.

The goal of *traditional* computational surface analysis is to find a representation of a surface that is expressive enough to capture details relevant for the problem or task at hand, while being computationally light-weight. However, the effectiveness of Convolutional Neural Network (CNN) in image processing opened new doors to surface processing. The design of efficient convolution-like operations to adapt neural networks (NN) to surfaces alleviated the need for complex and detailed representations, to the point where most state of the art architectures use extrinsic vertex coordinates as input, letting the NN models learn the surface structure at multiple scales. While some attempts were made to use well known representations as inputs, yielding some increase in performance [38, 35], the general consensus is that the model should be able to learn it by itself [25]. While it is true that neural networks are efficient at capturing surface features at multiple scales, the use of a local surface representation that is more expressive and more natural to interpret than extrinsic coordinates should naturally improve the performance of the network. The optimal choice of representation should be somewhere between coarse extrinsic vertex coordinates, and more complex representations.

38th Conference on Neural Information Processing Systems (NeurIPS 2024).

In Riemannian geometry, the *shape operator* is the main tool linking the intrinsic geometry of a surface with its bending and curving in ambient space. Its eigenvalues are the principal curvatures. Their product and evenly divided sum give precisely the Gauss and mean curvature of the surface, respectively. We hypothesize that the optimal choice for a local surface representation that meets the requirements of surface processing is the set of principal curvatures: they characterize the surface up to isometry (location and orientation in space); they are purely local, which allows the neural network to decide on more general surface features; and they are lightweight, leaving very little computational overhead in any scenario. This work tests our hypothesis against two widely used representations of surfaces in three state of the art NN architectures.

In the next section we give an overview of surface processing, and introduce shape representations and learning methods. In section 3 we give an introduction to the Shape Operator, although we should remark now that this is *not* the first appearance of the shape operator in the surface processing literature. The shape operator has already become an efficient tool in surface processing and is, among other things, used to define local tangent frames and compute surface *features* like creases [31]. Following these introductions, we then conduct extensive experiments in section 4, comparing principal curvature with three other representations in conjunction with three different NN architectures, on two segmentation datasets and one classification dataset, that shows how principal curvature enhances any state of the art model in different tasks. In addition to outperforming other methods, we show that this more concise representation is faster to compute, leaving minimal computational overhead when added to a pipeline.

## 2  Related work

The first step to surface processing is usually its discretisation as a mesh or point cloud, which is particularly useful for visualisation or rendering. From this starting point, novel representations have been derived in an effort to provide tools for different surface related tasks. These tasks include surface matching, semantic segmentation, classification, or even shape retrieval.

Historically, the general trend has been to find compact descriptors of a shape which could be then compared within a dataset. A long list of such descriptors exist, among them signature-based descriptors, such as Heat Kernel Signature (HKS) [42] or wave kernel signature [4], proved to be particularly efficient. Closely related are histogram methods, which are often combined with signatures to provide expressive representations such as the SHOT [37] or the Echo [26] descriptors. Geometric measure theory has also been a source of inspiration for developing efficient representations, such as geometric currents [5] leveraging on finite elements, or kernel-based currents [43] and varifolds [10] tailored for shape deformation. Such representations can be used in conjunction with classical statistical analysis tools, e.g [26, 37, 5], although they are often building blocks for specialised methods on surfaces such as LDDMM [49], functional maps [29] or spectral-based analysis [45].

With the advent of deep learning, many methods previously stated were re-written with the help of neural networks resulting in Deep functional maps [20], and ResNet-LDDMM [2], to name but a few. Representations of surfaces themselves were proposed as neural networks, such as DeepSDF [32] or DeepCurrents [30]. As convolutions proved particularly effective when learning on images, i.e structured grids, some work proposed voxel-based solutions to the study of surfaces [23]. Others suggested representing surfaces as geometric images [40], on which convolutions can be applied. A second generation of *geometric deep learning* has focused on building network architecture specifically tailored to work directly on surfaces, i.e meshes or point clouds. From a point cloud perspective, Point Net [34] and its extension Point Net ++ [35] consider the surface as a set of points by applying set operations on them. Among others, DGCNN [46] applies a convolution-like operation on *dynamic graphs* constructed layer-wise. MeshCNN [16] on the other hand fully leverages the mesh structure to develop operation unique to triangulations. Transformer based architectures have also appeared for the specific purpose of surface processing [18]. Among them, and in a similar vein as before, GaTr [8] proposes to represent geometric data in an algebra of choice and designs an architecture with operations belonging to this algebra. An effort to have efficient generalisation of convolutions on surfaces was proposed by [22], although the lack of global coordinates creates ambiguity in local operations. To alleviate this problem, a large body of work has proposed *rotation equivariant* operations, namely GemCNN [12], augmenting graph NNs, or field convolutions [25]. Finally, recent models propose to bypass the problem of generalising convolutions by focusing

on well defined operations on surfaces, such as discrete exterior calculus in Hodge Net [41], heat diffusion in Diffusion Net [38] or a suit of known operators in Delta Net [48].

As model architectures include more and more knowledge of shapes, the need for a better representation of the input to these models has decreased. Outside of models that contain operations proper to the structure of choice (e.g [16, 8]), most models naturally accept as input the coordinates at every point. Some papers propose to augment the model by inputing higher dimensional descriptors initially designed for a more *direct* analysis, such as the ones previously mentioned (e.g HKS, WKS, SHOT). Such proposals can be seen in [38], where HKS interacts well with the diffusion part of the architecture, or [35] were they combine HKS, WKS, Gaussian curvature through concatenation and PCA. However, recent work has dismissed this idea [25], citing the results of Diffusion Net [38] which show no great improvement when moving from coordinates to HKS.

As methods for learning on surfaces have evolved, we suggest that a better input representation is a simpler one, yet is more expressive than coordinates. We suggest that we can scale back to the simplest differentiable invariant of a surface: its curvature.

## 3   The Shape Operator

Here we give a conceptual introduction to the Shape Operator and describe how its eigenvalues completely characterize the surface to which it belongs (section 3.1). In section 3.2 we describe an explicit calculation of the Shape Operator which `igl`'s implementation of the principal curvatures is based on – this is the implementation we use in our experiments (see section 4.1). Those already familiar with the differential geometry of curves and surfaces may skip ahead to section 4; for others this section serves as a concise introduction – although, we do rely on a basic understanding of functions of several variables and their derivatives, and surfaces and their tangent spaces.

Surfaces in $\mathbb{R}^3$ will be denoted by $S$ and $\overline{S}$, points in surfaces by $p$'s and $q$'s, and the tangent space to $S$ at a point $p \in S$ by $T_pS$. Maps from $\mathbb{R}^3$ to itself will be denoted by $F : \mathbb{R}^3 \to \mathbb{R}^3$, and their derivatives at a point $p$ by $DF_p$ – the Jacobian matrix. A parameterisation $X$ of a smooth surface $S$ is a diffeomorphism between an open set $U \subset \mathbb{R}^2$ and an open set $V \subset S$, and provides a mathematical description of $S$ as it lies in $\mathbb{R}^3$. The standard Euclidean inner product on $\mathbb{R}^3$ will be signified by $\langle \cdot, \cdot \rangle$, and it's restriction to a surface $S$ and its tangent bundle $TS = \coprod_{p \in S} T_pS$ by $g_S(\cdot, \cdot)$, which we call the induced metric. A normal vector to $S$ at $p$ is one which is orthogonal to every vector $v$ in $T_pS$ (measured in $\langle \cdot, \cdot \rangle$) and will be denoted by $N_p$; if we have a field of normal vectors in an open set around $p$ then this field will be denoted simply by $N$.

The *Shape Operator* of a surface $S$ at a point $p \in S$ measures the rate at which surface normal vectors $N$ separate around $p$, which is precisely the bending of the surface in space:

**Definition 1** *Given a point $p$ on a surface $S \subset \mathbb{R}^3$, and the unit normal vector $N$ defined on a neighbourhood $U$ of $p$, the shape operator is the linear map*

$$\mathbf{S}_p : T_pS \longrightarrow T_pS$$
$$v \longmapsto -\nabla_v N,$$

*where $T_pS$ denotes the tangent space of $S$ at point $p$*

In other words, the shape operator $\mathbf{S}_p$ tells us how the normal vector changes as we move in $S$, in the direction of $v$ from $p$. One possible way to visualise the shape operator is through the Gauss map, which identifies each point $p \in S \subset \mathbb{R}^3$ with its unit normal vector $N_p$, now thought of as a point in $\mathbb{S}^2$. The shape operator is then the differential of the Gauss map at $p$ and is a tangent vector to $\mathbb{S}^2$ at the image $N_p$ of $p$, as illustrated in figure 1.

The operator $\mathbf{S}_p$ is linear for each $p \in S$, and self-adjoint in the Euclidean inner product $\langle \cdot, \cdot \rangle$:

$$\langle \mathbf{S}_p(v), w \rangle = \langle v, \mathbf{S}_p(w) \rangle. \tag{1}$$

It can therefore be represented by a symmetric $2 \times 2$ matrix $[\mathbf{S}_p] : T_pS \to T_pS$ at each point $p \in S$. It is well-known that symmetric matrices admit a complete system of orthonormal eigenvectors $(e_1, e_2)$ spanning the space on which they act. The matrix representation $[\mathbf{S}_p]$ with respect to the basis $(e_1, e_2)$ has the simple form:

$$[\mathbf{S}_p] = \begin{pmatrix} \kappa_1 & 0 \\ 0 & \kappa_2 \end{pmatrix}. \tag{2}$$

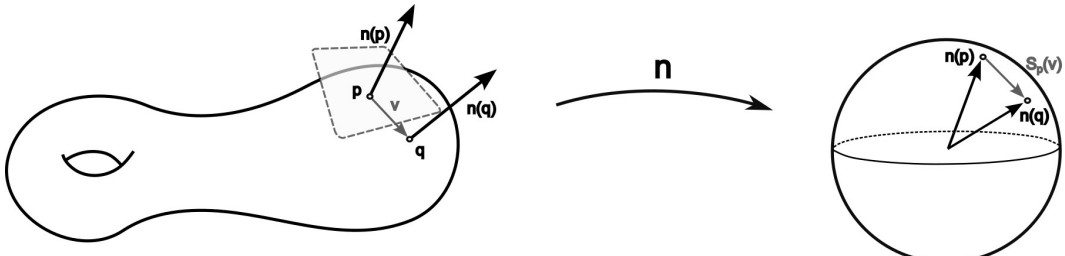

Figure 1: The shape operator may be visualised via the Gauss map.

Where $\kappa_1$ and $\kappa_2$ are the eigenvalues of $S_p$.

**Definition 2** *Let $S$ be a surface in $\mathbb{R}^3$, $p$ a point in $S$, $\mathbf{S}_p$ the shape operator at $p$ and $[\mathbf{S}_p]$ its matrix representation.*

1. *The eigenvalues $\kappa_1(p)$ and $\kappa_2(p)$ of $[\mathbf{S}_p]$ at $p$ are the* principal curvatures *of $S$ at $p$, and their corresponding eigenvectors $e_1$ and $e_2$ are the* principal directions*;*

2. *The* Gauss curvature $\kappa$ *of $S$ at $p$ is the product $\kappa_1(p) \cdot \kappa_2(p)$ of the principal curvatures;*

3. *The* mean curvature $H_p$ *is the average $\frac{\kappa_1(p)+\kappa_2(p)}{2}$ of the principal curvatures*

*The Gauss and mean curvatures can be equivalently interpreted as the determinant and half the trace of $[\mathbf{S}_p]$, respectively.*

The importance of these quantities is two-fold: (1) two surfaces differ only in location and orientation in space if and only if they have the same principal curvatures (Theorems 9.1 and 9.2 in [28]) – that is, the shape operator completely characterizes the shape of a surface; and (2) the Gauss and mean curvature generate all possible differential invariants of a surface (see Guggenheim [15], Olver [27]) – in particular, Gauss and mean curvature are fundamental characteristics of the shape of a surface, and the inclusion of the higher order invariants they generate into a representation could even improve the results shown here.

### 3.1 Congruence

To explain how Gauss and mean curvature completely describe the shape of a surface we need a few more definitions.

An *isometry* of $\mathbb{R}^3$ is a map $F : \mathbb{R}^3 \to \mathbb{R}^3$ whose differential preserves the angles between tangent vectors at every point of $\mathbb{R}^n$:

$$\langle v, w \rangle_p = \langle DF_p \cdot v, DF_p \cdot w \rangle_p, \quad \forall v, w \in T_p\mathbb{R}^3. \tag{3}$$

If $g_S$ is the Riemannian inner product induced on $TS$ by the Euclidean inner product $\langle \cdot, \cdot \rangle$ then an isometry between two surfaces is a map $\eta : S \to \overline{S}$ whose differential preserves the angles between tangent vectors to $S$:

$$g_S(v, w) = g_{\overline{S}}(D\eta \cdot v, D\eta \cdot w). \tag{4}$$

Every isometry $F$ of $\mathbb{R}^3$ restricts to an isometry of surfaces $F|_S = \eta : S \to F(S)$, but the converse need not be true, unless an additional hypothesis on the shape operators is satisfied.

Two surfaces $S$ and $\overline{S}$ are *congruent* if there exists an isometry $F : \mathbb{R}^3 \to \mathbb{R}^3$ such that $F(S) = \overline{S}$; that is, congruent surfaces are surfaces which differ only in their location and orientation in space. It is clear that the shape operators $\mathbf{S}$ and $\overline{\mathbf{S}}$ of two congruent surfaces are related by

$$DF_p \cdot \mathbf{S}_p(v) = \overline{\mathbf{S}}_{F(p)}\left(DF_p \cdot v\right), \quad \forall v \in T_pS; \tag{5}$$

in particular, the matrices $[\mathbf{S}_p]$ and $[\overline{\mathbf{S}}_p]$ are conjugate to one another via $[DF_p]$. As per Theorem 9.2 of [28], if there exists an isometry $\eta : S \to \overline{S}$ such that

$$D\eta_p \cdot \mathbf{S}_p(v) = \overline{\mathbf{S}}_{\eta(p)}\left(D\eta_p \cdot v\right), \quad \forall v \in T_pS, \tag{6}$$

i.e. such that the matrices $[\mathbf{S}_p]$ and $[\overline{\mathbf{S}}_p]$ are conjugate to one another via the matrix representation $[D\eta_p]$, then there exists an isometry $F : \mathbb{R}^3 \to \mathbb{R}^3$ such that $F|_S(S) = \eta(S) = \overline{S}$, and the two surfaces are congruent. The conclusion of this brief mathematical digression is that two congruent surfaces have the same intrinsic geometry and shape in space, and two surfaces with the same intrinsic geometry and shape in space are congruent. This is what is sought after when representing shapes with intrinsic quantities.

## 3.2 Discrete curvature

As well as being an important theoretical tool, curvature is a central notion in mesh processing. A large body of work has been dedicated to estimating its discrete counterpart. Among them, many methods propose to infer Gaussian curvature directly, such as in [24], or involve the use of geometric measure theory [13], as in [11, 17]. Interestingly, many efficient methods propose to first discretize the shape operator in order to compute the Gaussian curvature from it. This is done either directly on the mesh triangles, such as in [36], or by first locally fitting a function to the surface, and then computing explicitly the shape operator. To get a better feel for why this is a natural construction of the shape operator, consider a surface $S$, given a point $p \in S$. Then the surface around $p$ can be parameterized as $X(u, v)$ with $(u, v) \in \mathbb{R}^2$. The inner product at $T_pS$, also called the first fundamental form, is then given for any two tangent vectors $v, w$ by:

$$(v, w)_p = v^T \begin{pmatrix} E & F \\ F & G \end{pmatrix} w, \tag{7}$$

where $E = \langle \partial_u X, \partial_u X \rangle$, $F = \langle \partial_u X, \partial_v X \rangle$ and $G = \langle \partial_v X, \partial_v X \rangle$. And the surface normal at $p$ can be defined as

$$n = \frac{\partial_u X(u, v) \times \partial_v X(u, v)}{|\partial_u X(u, v) \times \partial_v X(u, v)|}. \tag{8}$$

We can now define the second partial derivatives of $X$ in the normal direction $n$, a quantity called the second fundamental form, noted $\mathbb{II}$:

$$\mathbb{II} = \begin{pmatrix} L & M \\ M & N \end{pmatrix} \tag{9}$$

where $L = \langle \partial_{uu} X, n \rangle$, $M = \langle \partial_{uv} X, n \rangle$, and $N = \langle \partial_{vv} X, n \rangle$. The partial derivatives of the surface normal can then be expressed via the Weingarten equations, in terms of the components of the first and second fundamental form:

$$\partial_u n = \frac{FM - GL}{EG - F^2} \partial_u X + \frac{FL - EM}{EG - F^2} \partial_v X$$
$$\partial_v n = \frac{FN - GM}{EG - F^2} \partial_u X + \frac{FM - EN}{EG - F^2} \partial_v X.$$

This enables us to write the matrix form of the shape operator at $p$ as:

$$[\mathbf{S}_p] = (EG - F^2)^{-1} \begin{pmatrix} LG - MF & ME - LF \\ ME - LF & NE - MF \end{pmatrix} \tag{10}$$

From these derivations, it becomes interesting to find good local parametrisation of surfaces, that is, a bi-variate scalar function $f$ such that:

$$X(u, v) = (u, v, f(u, v)) \tag{11}$$

The shape operator can then be easily derived from the first and second derivatives of $f$. An efficient way to find such functions is via osculating jets, proposed in [9]. For the following experiments, we use a multi-scale version of this, proposed in [31], in which the shape operator is computed by using neighbourhoods of varying size around a point, yielding a robust method for estimating curvature on a mesh.

## 4 Experiments

We test the representation of surfaces by the principal curvatures $\kappa_1, \kappa_2$ and Gaussian curvature $\kappa$ against the three most commonly used representations: the **HKS** [42], the **SHOT** descriptor [37], and

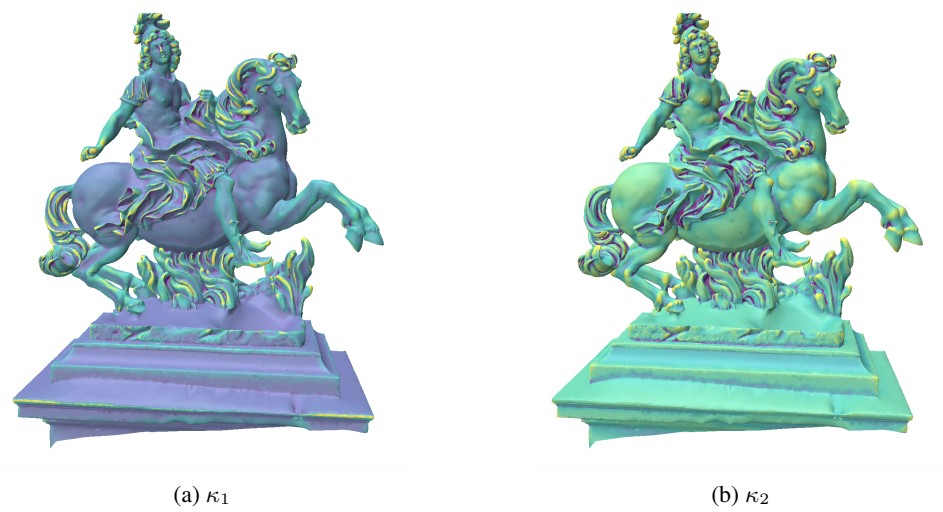

(a) $\kappa_1$          (b) $\kappa_2$

Figure 2: Principal curvature visualisation of a Louis XIV statue.

the **extrinsic coordinates**. The HKS is a purely intrinsic representation derived from the Laplace operator, and constitutes the most widely used signature-based method to represent shapes. The SHOT representation is a descriptor mixing signature and histogram-based methods to describe shapes, and is therefore an extrinsic representation. As they belong to two different classes of surface representations we believe they are the most adequate for benchmarking our proposed curvature representation.

All representations are tested with three different architectures. We regard **Diffusion Net** [38] as the state of the art in NN architectures, as it shows the most promising results on general benchmark tasks. In addition, it shows very little difference in performance when changing the input from coordinates to HKS, making it the *hardest* test for our representation. **Point Net ++** [35] has been designed as a general method to process shapes arising in many situations, including controlled environments – as in our case – but also from segmented images encountered in the autonomous driving field [35]. As such **Point Net ++** uses the least geometric structure to describe a surface: all one needs is a point set. We believe that in this case, using better surface information for the input will greatly enhance the performance of the model. The authors of PointNet++ have already touched on this subject, recommending a linear combination of HKS, WKS and Gaussian curvature, followed by a PCA projection, leading to a 64 dimensional feature per point. We aim to show that a 2d (or even 1d) input of curvature information is more relevant for a smaller computational cost. **Delta Net** [48] proposes an architecture intrinsic to surfaces by design, by combining four operators defined on the surface: Laplacian, divergence, curl and norm. Most papers that propose other surface descriptors rather than coordinates as input, do it solely to have an intrinsic representation of the surface. Curvature gives isometry invariance (section 3), and we further believe it is also more robust, numerically. Better performance from a curvature based representation in this architecture would support this belief.

Finally, we pick three tasks of varying complexity to measure the impact of each method: human segmentation [21], molecular segmentation [6], and shape classification [19]. Examples from each dataset are shown in figure 3.

## 4.1 Implementation

The performance of each representation is strongly dependent on the chosen implementation. We have tried to be as fair as possible by not developing our own implementations of existing work and instead using implementations which have already been tried, tested, and validated in the literature. For calculating the discrete principal curvatures via quadratic surface fitting, we have used `igl`'s implementation with a fixed neighbourhood radius of 5; the Gaussian curvature $\kappa$ is then computed directly as the product of $\kappa_1, \kappa_2$. HKS depends on the Laplacian, and we have used the method implemented in `robust-laplacian` based on [39] - this is also consistent with what is used in Diffusion Net. The eigendecomposition of the Laplacian is then performed with `scipy`. For the

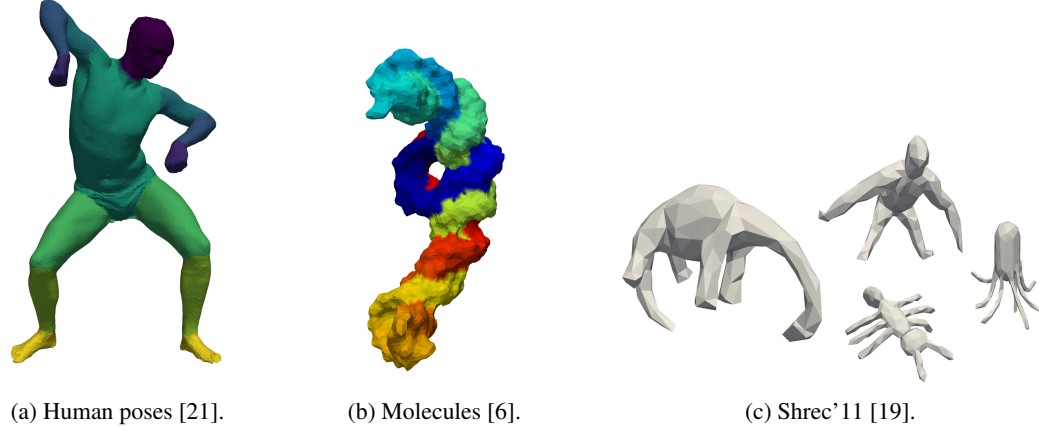

(a) Human poses [21]. (b) Molecules [6]. (c) Shrec'11 [19].

Figure 3: Samples of the segmentation and classification datasets used for experiments.

SHOT representation, we use the implementation in the `pcl` library, which computes 352 features per vertex, in this case we normalise all shapes and use a ball of radius .1; all other parameters are left untouched.

Regarding the neural networks, we use the implementations made publicly available by the authors, modifying only when needed to accommodate more than just coordinates as input. We also use the same parameters proposed in each paper when they are known, which we detail for every task below. We make all our code and experiments available at `https://github.com/Inria-Asclepios/shape-nets`

## 4.2 Time Complexity

As a first experiment, we compute[1] for each representation method, the computation time as a function of the number of points in a surface. The performances are reported in figure 4. HKS and curvature are both efficient for meshes with up to 100k points. However, curvature is consistently faster, even for larger meshes (up to 500k points), displaying the very small overhead incurred by the use of curvature in surface models.

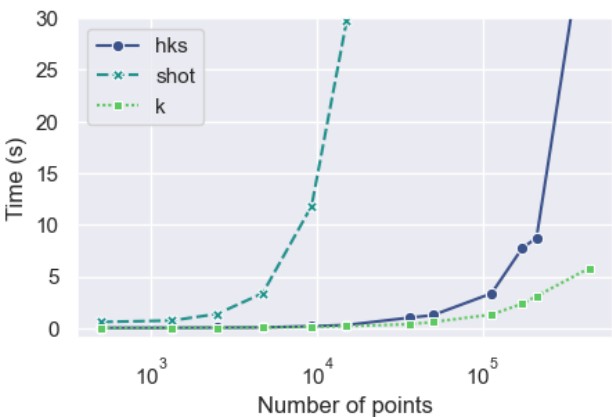

Figure 4: Time of computation for each representation with respect to the number of points in a mesh.

---

[1]Computed on an Apple M2 chip

### 4.3 Human Anatomy segmentation

We first segment the human parts from the composite dataset proposed in [21], containing samples from other human dataset, namely FAUST [7], SCAPE [3], Adobe [1], MIT [44], and SHREC07 [14]. As in the original paper, we use the SHREC07 dataset as test set. Similar to [47], we differ from [21] by evaluating on vertices rather than faces. For Point Net ++ and Delta Net we resample each shape to 1024 points, and we leave the meshes untouched for Diffusion Net, as per the experiments conducted in each paper. We optimise the negative log-likelihood for 100 epochs, with the ADAM optimiser and a scheduler step every 20 epochs. We ran the experiment 5 times and have reported the mean test accuracy in table 1.

|  | $xyz$ | $\text{shot}_{16}$ | $\text{shot}_{64}$ | hks | $\kappa_1, \kappa_2$ | $\kappa$ |
|---|---|---|---|---|---|---|
| Point Net ++ | 69.6 | 71.4 | 72.4 | 78.1 | **80.6** | 74.5 |
| Delta Net | 72.4 | 58.1 | 66.2 | 68.9 | **86.8** | 60.0 |
| Diffusion Net | 94.7 | 95.0 | 95.0 | 95.1 | **97.4** | 95.4 |

Table 1: Test accuracies (%) on the Human part segmentation task.

The results highlight the assumption that better representations lead to better performance. PointNet++ showed the greatest improvement when moving away from coordinates: this is due to its architecture having the least amount of geometric information at baseline. The better results come from the principal curvatures, and show how expressive this representation is. The effects of the principal curvatures are even more pronounced in the Delta Net experiment, where $\kappa_1, \kappa_2$ greatly outperform all other methods. It's interesting to note that in this experiment the coordinate representation performs better than the other more complex representations. Diffusion Net may show that it is more robust to the type of input, as long as it loosely describes the shape, however the improvement brought by the principal curvature is still significant. To further demonstrate the impact of a good representation, even in the case of Diffusion Net, we show in 5 the worst cases for $xyz$ and $\kappa_1, \kappa_2$ inputs. The clear improvement in this case may be even more important than general accuracy in some cases, e.g with human-in-the-loop type corrections.

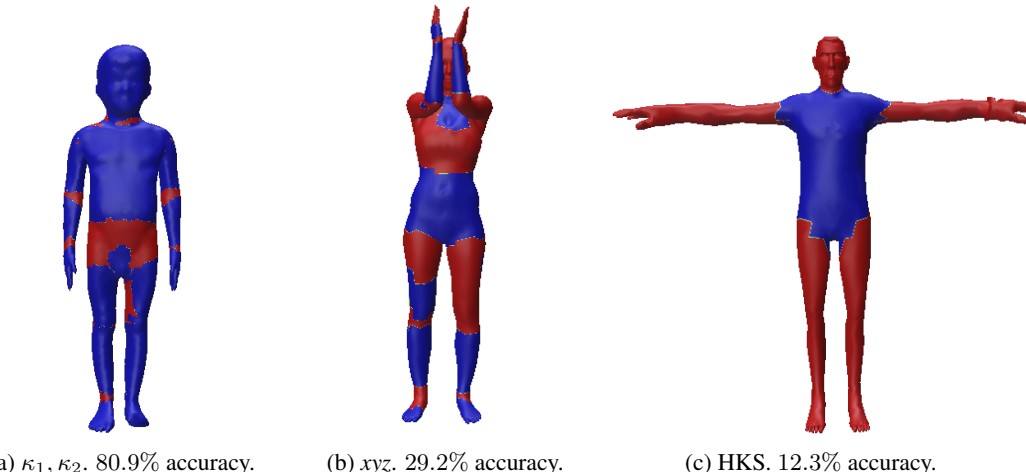

(a) $\kappa_1, \kappa_2$. 80.9% accuracy.     (b) $xyz$. 29.2% accuracy.     (c) HKS. 12.3% accuracy.

Figure 5: Human part segmentation with Diffusion Net. Worst cases for different representations, blue shows the correct prediction, red the error.

### 4.4 Molecular segmentation

The molecular dataset made available by [6] and first proposed in [33], can be considered a harder segmentation task then the Human part dataset: it proposes a wider range of shapes in the form of RNA molecules, and a 260-way part segmentation task. We resample all meshes to 2048 points, except in the case of Diffusion Net where we kept the original discretisation. We evaluate all our baselines on 5 random splits with a train-test ratio of 80-20. We run the models for 200 epochs, and report the mean test accuracies in table 2.

|               | $xyz$ | $\text{shot}_{16}$ | $\text{shot}_{64}$ | hks  | $\kappa_1, \kappa_2$ | $\kappa$ |
|---------------|-------|--------------------|--------------------|------|----------------------|----------|
| Point Net ++  | 35.4  | 70.9               | 71.9               | 70.2 | **72.4**             | 69.0     |
| Delta Net     | 29.2  | 45.6               | **56.5**           | 49.6 | 55.5                 | 29.2     |
| Diffusion Net | 82.6  | 88.4               | 89.1               | 85.6 | **89.4**             | 84.0     |

Table 2: Test accuracies (%) on the Molecular segmentation task.

Again, we see a significant improvement when using a better representation of the surface in the case of PointNet++, going from failing in the case of coordinates to outperforming Delta Net – with principal curvatures giving the best performances. Diffusion Net shows a non-negligable jump in performance as well. Although the SHOT descriptor outperforms other representations in the case of Delta Net, the general performance of this architecture is underwhelming. We believe this is due to the accumulation of errors in the discretisation of surface operators used. Indeed, one layer computes a chain of 6 operators on the surface: since the RNA shapes are very irregular, the error for each operator could be significant.

## 4.5 Classification

In addition to segmentation tasks, we propose to compare representations in the context of classification. This experiment should show whether or not geometrically informative inputs interact well with pooling-type operations. We choose the widely adopted baseline Shrec11, proposed in [19]. It is a 30-way classification dataset with 20 shapes per category. We choose the simplified mesh dataset and the harder version of training, using only 10 samples per class and evaluating on the test. We perform our experiments on 5 random splits. We train our baselines for 100 epochs with a scheduler step at epoch 50 and optimise the cross-entropy loss with a label smoothing factor of $0.2$. Resulting mean test accuracies are shown in table 3.

|               | $xyz$ | $\text{shot}_{16}$ | $\text{shot}_{64}$ | hks  | $\kappa_1, \kappa_2$ | $\kappa$ |
|---------------|-------|--------------------|--------------------|------|----------------------|----------|
| Point Net ++  | 71.5  | 69.8               | 60.7               | 60.8 | 85.7                 | **96.2** |
| Delta Net     | 75.7  | 54.9               | 60.4               | 98.6 | 90.1                 | **98.8** |
| Diffusion Net | 80.3  | 52.6               | 67.4               | 98.9 | 94.2                 | **99.1** |

Table 3: Test accuracies (%) on the Shrec11 classification task.

Yet again we observe a significant improvement when turning to better representations, even more so when using Gaussian curvature $\kappa$. Additionally, figure 6 shows that all geometric representations yield less variability across each folds. In addition, HKS, Gaussian curvature $\kappa$, and principal curvature $\kappa_1, \kappa_2$ converge much faster than all others.

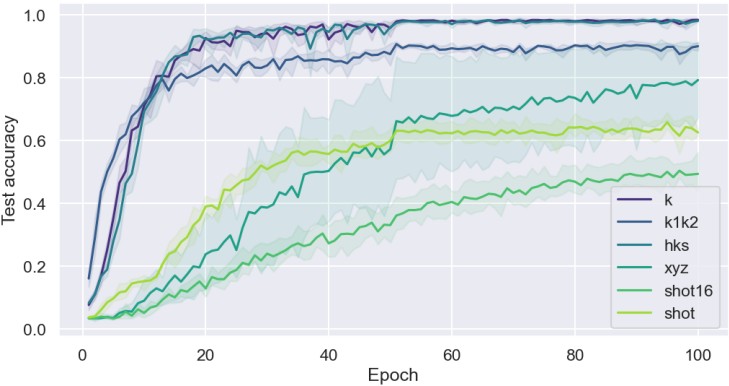

Figure 6: Evolution of the test accuracy with $95\%$ confidence interval by epochs per representations across folds, for the Shrec07 dataset using Diffusion Net.

The fact that gaussian curvature, closely followed by HKS outperform principal curvature in this classification task seems to indicate that Gaussian curvature interacts better with pooling operations present in classification architectures. Interestingly, all three architectures tested here have different

ways of performing the pooling operation. Although it is hard to give any analytical reasoning to this behavior, we believe it is simply the fact that gaussian curvature is already an aggregation of the principal curvatures, that it shows better performance in classification tasks.

For each experiment, additional metrics can be found in Appendix A.1.

## 4.6 Noisy data

We propose one final experiment to highlight the robustness of input features to noisy data. We focus on three representations: HKS, known to be robust to noise as it computes a representation at multiple scale; extrinsic coordinates that are directly impacted by the noise; and principal curvatures, known to be less robust to noise as a purely local descriptor. To compare these representations we pick the diffusion net trained on the human pose dataset, and we add noise to the dataset at inference time. Specifically, we add gaussian noise with a standard deviation of $1\%, 3\%, 5\%, 7\%$ and $10\%$ of the diagonal length of the bounding box of each shape. Examples of the noisy data can be seen in Appendix A.2. Results show that the accuracy for all three features worsen at the same rate, as shown in figure 7, showing that principal curvature can be a viable choice even in the presence of noisy data.

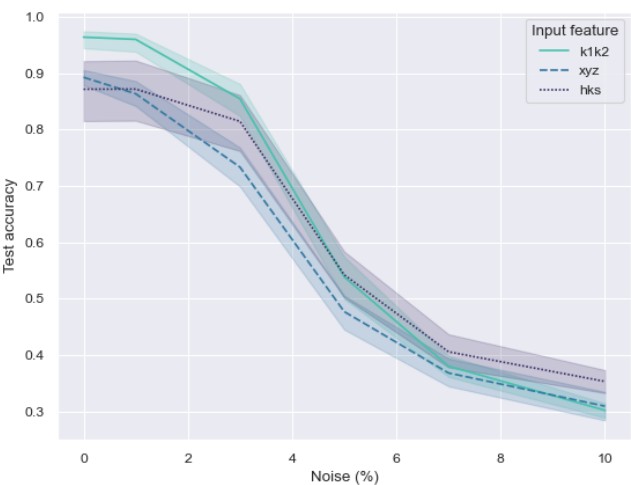

Figure 7: Evolution of the test accuracy on the human pose segmentation task for inputs $(k_1, k_2)$, HKS and the extrinsic coordinates when noise is added to the shapes.

## 5 Conclusion

In this work we have shown that curvature should be the representation of choice when it comes to processing surfaces with neural networks. In almost all experiments the principal and Gaussian curvatures performed better than any other choice of input, both qualitatively and quantitatively. In particular, this representation can be obtained with minimal computational overhead. Its combination with PointNet++, the architecture that has the least prior information about the surface, showed that it can help the network better understand the surface structure. When combined with Delta Net, which contains only intrinsic operations, the improvement indicates that curvature gives more than just a rigid transformation invariance. Even in the case of Diffusion Net, where the diffusion operation seems to interact nicely with any representation, curvature as input showed significant amelioration. For these reasons, we believe curvature should become the standard practice when using models to learn on surfaces. Finally, although experiments have shown that gaussian curvature outperforms principal curvatures on classification tasks, we would like to further define those guidelines in future work, as well as compare representations in a wider range of tasks and architectures.

# 6 Acknowledgements

This Work has been funded by PARIS - ERACoSysMed grant number 15087, by G-Statistics - ERC grant number 786854 and has been supported by the French government through the National Research Agency (ANR) Investments in the 3IA Côte d'Azur (ANR-19-P3IA-000). The authors are grateful to the OPAL infrastructure from Université Côte d'Azur for providing computing resources and support.

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

# A Appendix

## A.1 Segmentation and classification detailed results

We present below the complete results for each experiment. For each dataset, and each neural network architecture, we show the Accuracy, Balanced accuracy, F1 score, and Specificity. They were all measured using 5-fold cross validation, and we give the results on the test sets in the form of *mean ± standard deviation*.

|  | Accuracy | Balanced accuracy | F1 score | Specificity |
|---|---|---|---|---|
| $xyz$ | $.696\pm.306$ | $.804\pm.393$ | $.662\pm.293$ | $.957\pm.49$ |
| $shot_{16}$ | $.714\pm.061$ | $.815\pm.15$ | $.689\pm.061$ | $.959\pm.234$ |
| $shot_{64}$ | $.724\pm.034$ | $.827\pm.13$ | $.693\pm.035$ | $.961\pm.201$ |
| hks | $\mathbf{.781}\pm.07$ | $.865\pm.159$ | $.754\pm.08$ | $.969\pm.203$ |
| $\kappa_1, \kappa_2$ | $\mathbf{.806}\pm.079$ | $.871\pm.14$ | $.799\pm.089$ | $.972\pm.196$ |
| $\kappa$ | $\mathbf{.745}\pm.165$ | $.826\pm.227$ | $.714\pm.148$ | $.964\pm.32$ |

Table 4: **Human pose segmentation** - Point Net ++ results.

|  | Accuracy | Balanced accuracy | F1 score | Specificity |
|---|---|---|---|---|
| $xyz$ | $.724\pm.265$ | $.223\pm.076$ | $.223\pm.076$ | $.724\pm.265$ |
| $shot_{16}$ | $.581\pm.095$ | $.215\pm.05$ | $.215\pm.05$ | $.581\pm.095$ |
| $shot_{64}$ | $.662\pm.075$ | $\mathbf{.302}\pm.084$ | $\mathbf{.302}\pm.084$ | $.662\pm.075$ |
| hks | $.689\pm.14$ | $\mathbf{.303}\pm.043$ | $\mathbf{.303}\pm.043$ | $.689\pm.14$ |
| $\kappa_1, \kappa_2$ | $\mathbf{.868}\pm.128$ | $\mathbf{.299}\pm.114$ | $\mathbf{.299}\pm.114$ | $\mathbf{.868}\pm.128$ |
| $\kappa$ | $.6\pm.087$ | $.163\pm.054$ | $.163\pm.054$ | $.6\pm.087$ |

Table 5: **Human pose segmentation** - Delta Net results.

|  | Accuracy | Balanced accuracy | F1 score | Specificity |
|---|---|---|---|---|
| $xyz$ | $.947\pm.013$ | $.943\pm.015$ | $.943\pm.015$ | $.947\pm.013$ |
| $shot_{16}$ | $.95\pm.017$ | $.946\pm.018$ | $.946\pm.018$ | $.95\pm.017$ |
| $shot_{64}$ | $.95\pm.018$ | $.944\pm.021$ | $.944\pm.021$ | $.95\pm.018$ |
| hks | $.951\pm.04$ | $\mathbf{.969}\pm.055$ | $.947\pm.039$ | $\mathbf{.993}\pm.07$ |
| $\kappa_1, \kappa_2$ | $\mathbf{.975}\pm.014$ | $\mathbf{.971}\pm.015$ | $\mathbf{.971}\pm.015$ | $.975\pm.014$ |
| $\kappa$ | $.954\pm.014$ | $.951\pm.016$ | $.951\pm.016$ | $.954\pm.014$ |

Table 6: **Human pose segmentation** - Diffusion Net results.

|  | Accuracy | Balanced accuracy | F1 score | Specificity |
|---|---|---|---|---|
| $xyz$ | $.354\pm.008$ | $.202\pm.001$ | $.202\pm.001$ | $.354\pm.008$ |
| $shot_{16}$ | $.709\pm.004$ | $.592\pm.016$ | $.592\pm.016$ | $.709\pm.004$ |
| $shot_{64}$ | $\mathbf{.719}\pm.009$ | $\mathbf{.602}\pm.01$ | $\mathbf{.602}\pm.01$ | $\mathbf{.719}\pm.009$ |
| hks | $.703\pm.017$ | $.574\pm.003$ | $.574\pm.003$ | $.703\pm.017$ |
| $\kappa_1, \kappa_2$ | $\mathbf{.724}\pm.013$ | $.597\pm.013$ | $.597\pm.013$ | $\mathbf{.724}\pm.013$ |
| $\kappa$ | $.69\pm.008$ | $.572\pm.008$ | $.572\pm.008$ | $.69\pm.008$ |

Table 7: **RNA molecules segmentation** - PointNet++ results.

|  | Accuracy | Balanced accuracy | F1 score | Specificity |
|---|---|---|---|---|
| $xyz$ | $.292\pm.018$ | $.182\pm.012$ | $.182\pm.012$ | $.292\pm.018$ |
| $shot_{16}$ | $.456\pm.008$ | $.352\pm.016$ | $.352\pm.016$ | $.456\pm.008$ |
| $shot_{64}$ | $\mathbf{.565}\pm.010$ | $.299\pm.008$ | $.299\pm.008$ | $\mathbf{.565}\pm.010$ |
| hks | $.496\pm.017$ | $.392\pm.008$ | $.392\pm.008$ | $.496\pm.017$ |
| $\kappa_1, \kappa_2$ | $\mathbf{.555}\pm.023$ | $\mathbf{.496}\pm.013$ | $\mathbf{.496}\pm.013$ | $\mathbf{.555}\pm.023$ |
| $\kappa$ | $.292\pm.008$ | $.142\pm.008$ | $.142\pm.008$ | $.292\pm.008$ |

Table 8: **RNA molecules segmentation** - Delta Net results.

|  | Accuracy | Balanced accuracy | F1 score | Specificity |
|---|---|---|---|---|
| $xyz$ | .826±.001 | **.851**±.106 | .704±.003 | **.999**±.129 |
| shot$_{16}$ | .884±.002 | .765±.008 | **.765**±.008 | .874±.002 |
| shot$_{64}$ | **.891**±.001 | **.781**±.008 | .781±.008 | .879±.001 |
| hks | .856±.596 | **.873**±.622 | .758±.536 | **.999**±.697 |
| $\kappa_1, \kappa_2$ | **.894**±.008 | **.873**±.064 | **.783**±.025 | **.999**±.082 |
| $\kappa$ | .84±.008 | **.862**±.113 | .718±.011 | **.999**±.12 |

Table 9: **RNA molecules segmentation** - Diffusion Net results.

|  | Accuracy | Balanced accuracy | F1 score | Specificity |
|---|---|---|---|---|
| $xyz$ | .715±.037 | .852±.019 | .709±.034 | .99±.001 |
| shot$_{16}$ | .698±.035 | .844±.018 | .694±.036 | .99±.001 |
| shot$_{64}$ | .607±.031 | .797±.016 | .594±.038 | .986±.001 |
| hks | .608±.181 | .797±.093 | .602±.184 | .986±.006 |
| $\kappa_1, \kappa_2$ | .857±.007 | .926±.004 | .852±.007 | .995±.0 |
| $\kappa$ | **.962**±.011 | **.98**±.006 | **.961**±.011 | **.999**±.0 |

Table 10: **Shrec classification** - Point Net ++ results.

|  | Accuracy | Balanced accuracy | F1 score | Specificity |
|---|---|---|---|---|
| $xyz$ | .757±.027 | .753±.031 | .753±.031 | .757±.028 |
| shot$_{16}$ | .549±.029 | .533±.026 | .532±.026 | .549±.029 |
| shot$_{64}$ | .604±.026 | .597±.019 | .597±.019 | .604±.026 |
| hks | **.986**±.005 | **.985**±.006 | **.986**±.006 | **.986**±.005 |
| $\kappa_1, \kappa_2$ | .887±.020 | .881±.019 | .881±.019 | .887±.020 |
| $\kappa$ | **.988**±.004 | **.988**±.004 | **.988**±.004 | **.988**±.004 |

Table 11: **Shrec classification** - Delta Net results.

|  | Accuracy | Balanced accuracy | F1 score | Specificity |
|---|---|---|---|---|
| $xyz$ | .803±.159 | .898±.082 | .791±.169 | .993±.005 |
| shot$_{16}$ | .526±.054 | .755±.028 | .513±.053 | .984±.002 |
| shot$_{64}$ | .674±.033 | .831±.017 | .668±.036 | .989±.001 |
| hks | **.989**±.006 | **.994**±.003 | **.989**±.006 | **1.0**±.0 |
| $\kappa_1, \kappa_2$ | .922±.009 | .96±.004 | .919±.008 | .997±.0 |
| $\kappa$ | **.991**±.003 | **.995**±.001 | **.991**±.003 | **1.0**±.0 |

Table 12: **Shrec classification** - Diffusion Net results.

## A.2 Noisy data exemples

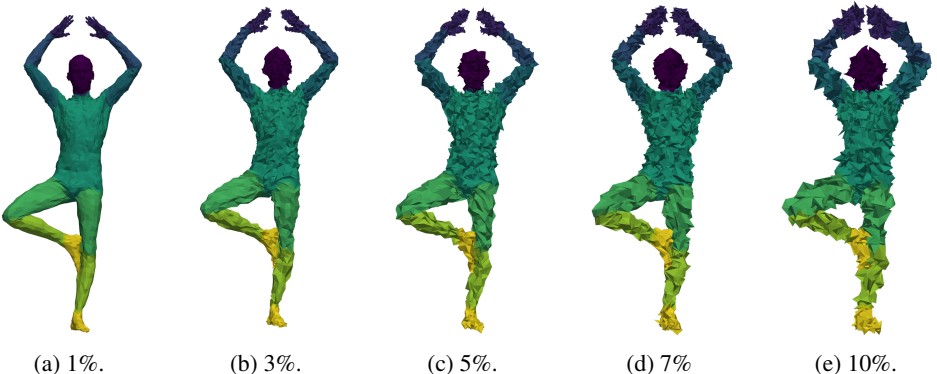

(a) 1%.      (b) 3%.      (c) 5%.      (d) 7%      (e) 10%.

Figure 8: Different quantity of noise added to a shape from the human pose dataset, from 1% to 10% of the diagonal of the bounding box of the shape.

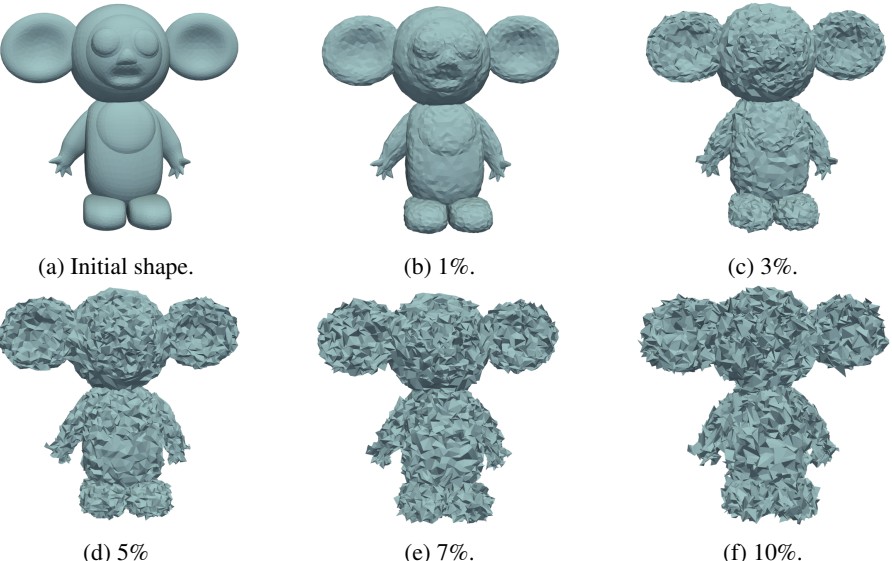

(a) Initial shape.      (b) 1%.      (c) 3%.

(d) 5%      (e) 7%.      (f) 10%.

Figure 9: Different quantity of noise added to a shape, from 1% to 10% of the diagonal of the bounding box of the shape.

