# OpenReview forum: "Improving Neural Network Surface Processing with Principal Curvatures"
_NeurIPS.cc/2024/Conference — NeurIPS 2024 poster_

### Official Review · Reviewer_ifJ1 · 2024-07-05

**Soundness:** 2
**Presentation:** 2
**Contribution:** 3
**Rating:** 6
**Confidence:** 3

**Summary:**

The paper promote to use the principal curvature as the surface presentation that can be better used in the modern neural network architectures. To support their hypothesis, the paper compares three different representations: HKS, SHOT descriptor and extrinsic coordinates in the use by PointNet++, Delta Net and Diffusion Net. Among all the providing experiment figures, principal curvatures provides the best performance.

**Strengths:**

- The paper provides an overview of the classic surface representations, it also provides an overview of the popular achitectures for geometric deep learning in the exsiting body of work.
- The paper work on a relatively meaningful topics in geometric deep learning, as it analysis different surface representations to be used by some popular surface-processing architectures.

**Weaknesses:**

- Quality:
  - Although section 3 provides the way to calculate the principal curvature, it takes up too much space but does not adding equivalently significant value to the paper, and thus appear to be a bit redundant.
  - The mathematical presentation in the paper is quit disorganized. Most of the formulation are not indexed. The definitions are given without a term [line 104, line 118].

- Incompliteness:
  -  Section 6 Acknowledgement is empty, which becomes another major drawback regarding to the presentation quality.

- Limited Contributions:
  -  The surface representations being analyzed in the paper is relatively limited. The task and the methods are not new, not does the paper provide an intelligence-stimulated combination of the existing method.

**Questions:**

Refer to the weaknesses.

**Limitations:**

Refer to the weaknesses.

---

> ### Author Rebuttal · Authors · 2024-08-07
>
> We would like to thank the reviewer for the time they have spent on reviewing our work.
>
> However, we are puzzled by several statements they have made, making it difficult to give a meaningful rebuttal. We will respond, point by point, as best as we can:
>
> > "... [section 3] takes up too much space but does not adding equivalently significant value to the paper, and thus appears to be a bit redundant."
>
> This is a value judgement and not a scientific judgement. Since curvature and the shape operator are the main tools used in the work, it is expected to state and define what these tools are, and give references. We believe this is normal scientific practice. If the reviewer believes otherwise, we ask they be more specific about what they expect or like to see instead.
>
> >"The mathematical presentation is quit disorganised. Most of the formulation are not indexed. The definitions are given without a term"
>
> The mathematical presentation is confined to two and a half pages, and is a little more than a sequence of definitions introducing the main mathematical objects we use. The definitions are clearly labelled, and since we make no explicit use of any particular formulas in later sections there is no real need to number them. We feel the reviewer is voicing, once again, a value judgement, which can be interpreted as deliberately obstructive and combative.
>
> > "Section 6 Acknowledgements is empty, which becomes another major drawback regarding to the presentation quality."
>
> This is perhaps the most puzzling statement made by the reviewer. This being a blind peer review process, we believe it is of common knowledge that acknowledgements should be left empty at this stage, to avoid compromising our anonymity. Using this argument for saying our paper is incomplete seems obstructive and combative.
>
> > "The surface representation being analyzed in the paper is relatively limited. The task and the methods are not new, not does the paper provide an intelligence stimulated combination of the existing methods"
>
> The belief that the analyzed representations (i.e. data) is limited is a value judgement, not a scientific judgement. The work addresses shape data which is used in numerous broad application domains, including  medical  imaging,  biology, architectural artifacts, etc. We have already addressed the point of novelty in our general answer, however we do not understand the problem raised by the reviewer with the tasks themselves not being new. They are commonly used as state-of-the-art benchmarking tasks, appropriate for comparing methods, with completely open sourced data, and varying levels of difficulty. We are unsure what you mean by the final clause in the second statement.

---

> ### Comment · Area_Chair_iouh · 2024-08-11
> **Additional comments for the review?**
>
> Dear ifJ1.
>
> The end of the discussion period is close.
> It looks like your review contains many general comments concerning the structure of the text, but not about the technical properties of the proposed method. Could you please provide in more detail any additional technical comments concerning the proposed method itself?

---

> ### Comment · Reviewer_ifJ1 · 2024-08-12
>
> ## I.Regardsing to the novelty...
> I had spent quit a long period in reading this paper, and trying to understand the ideas it convey. I really appologies that makes the authors think the review does not contribute much to a constructive discussion, it is also because the shape descriptor appears to be an unfamiliar topics to me. During the whole reviewing period, I keep thinking about how principal curvature contribute to the geometric learning scheme. There has been intensive research upon mesh and point cloud laplacian [1][2], mentioned about laplacian beltrami operator via its descrite forms of matrices contains important curvature, angles information. Also, a concurrent work "Manifold Diffusion Fields"[3] use the first k eigen vectors of the laplacian beltrami operator as intrinsic coordinates to replace x,y,z on mesh vertices or point cloud. They used it as positional encoder and use a transformer directly, got great improvement. These line of research are also conluded as the spectral methods in geometric deep learning, ref [4] (Section V)
>
> Another line of geometric deep learning on processing and learning the manifold surface, utilising the tangent bundle of surfaces. One of the most recent work in this line of research is [5]. Although they're task is to learning a texture genertion on single mesh, surface geometry are still being learned and can also be generalized to mesh deformation task.
>
> It could because my knowledge base are dominant by these evidence, that I did not fully recognise the contribution by the submission 17689. It appears to me that the paper performs experiment with the use of two surface representations with different neural networks. I recognized the overall experiments that being conducted in the paper, and the effort made by the authos, but think that the contribution is limited, also a major part in section 3 being vagues and hinder me to follow the paper. When the authors stating they are filling the research gaps, I was more expecting to see that the aforementioned research fields being addressed. Also it could also because my limited knowledge in the surface processing or shape operators, that did not realize the significance as much as authors and reviewer qKsN.
>
> - [1] Sorkine, Olga. "Laplacian mesh processing." Eurographics (State of the Art Reports) 4.4 (2005): 1.
> - [2] Lévy, Bruno, and Hao Zhang. "Spectral mesh processing." ACM SIGGRAPH 2010 Courses. 2010. 1-312.
> - [3] Elhag, Ahmed AA, et al. "Manifold Diffusion Fields." The Twelfth International Conference on Learning Representations.(2023).
> - [4] Bronstein, Michael M., Joan Bruna, Yann LeCun, Arthur Szlam, and Pierre Vandergheynst. "Geometric deep learning: going beyond Euclidean data." arXiv preprint arXiv:1611.08097 (2016).
> - [5] Mitchel, Thomas W., Carlos Esteves, and Ameesh Makadia. "Single Mesh Diffusion Models with Field Latents for Texture Generation." Proceedings of the IEEE/CVF Conference on Computer Vision and Pattern Recognition. 2024.
>
> ## II. Confusing points in the paper, especially section 3 Operator.
>
> Carrying with the questions, I kept reading the paper's method part, and hope to develop a better understanding.
> - (1) Two definitions are given without a terms, so I guess definition 1 (line 104) defines the shape operator,  definition 2 (line 118) defines the principal curvatures. If so, it would be better if the authors could root the definitions with some references.
> - (2) It is also confusing when the paper mentioned (e1,e2) in kune 115, what are their relationships to (k1,k2), the paper does not mention clearly, leaving an not rigorous impression to this part.
> - (3) line 137, (as the equation index not given), I do not find a explaintion to the operator DF that being used in the innert product <.,.>, so as the subscript p
> - (4) line 138, what is TS, the tangent bundel on surface S? Also S_hat is not proporly introduced.
> - (5) Based on (3), (4), the equations after line 140 is vague in the meaning, again the equations not indexed is a big problem. Does g_s refers to gaussian curvatures? What it g_s(.), definition or description not clearly given/easily found.
> - (6) line 139, to tell the difference between \ita and F, requires a prior knowledge in eulcidean space and manifold when reading.
>
> In a nutshell, section 3, especially section 3.1 is not rigourous in writting.
>
> ## III. Fianally...
> After reading the reviews and reply from reviewer qKsN, I realise that I am not fully recognise the contribution the submission 17689 has made to the surface processing and I was not very confidence about my understanding to the paper, while I also agree with reviewer LGgy with the weaknesses. Also the paper's writting for section 3 still being a big confusion to me, this could either be my rather limited understanding, or the paper need a revision to have a more rigorous way to better informed the audience. Thus, I will adjust my rating to boderline reject and lower down my confidence scroes as dicussed.

---

> > ### Author Response · Authors · 2024-08-14
> >
> > We thank the reviewer for this additional feedback. Their concerns now appear clearer although we might have misunderstood some claims. We are happy to address them below:
> >
> > Novelty and contributions of the work:
> > - ⁠Laplace mesh processing has been addressed in the paper; in fact, the HKS representation is the most successful representation stemming from this line of work, and in our paper we show that the principal, Gaussian, and mean curvatures outperform the HKS representation.
> > - While the Laplacian is linked to curvature (albeit through an equality relating two 2nd order differential operators), we are specifically suggesting in our work that curvature alone is a better suited input to neural networks processing shapes, which is strongly backed by our experiments. To the best of our knowledge, this has never been addressed in existing literature. Therefore (this could be a misunderstanding from our part) we do not agree with the claim that our work has entirely been done in the realm of spectral mesh processing. On the contrary, we believe our work is novel and contributes to the field of shapes processing.
> > - We thank the reviewer for the references they have provided. We will be happy to extend the contextualization of our work by adding a paragraph on transformer-based methods and including the suggested references.
> >
> > Notation and the mathematical aspect of the paper:
> > - The concerns of the reviewer about notations are now clearer. We will be happy to address them here (and in the revision of our paper) to facilitate the understanding of the mathematical aspects of our paper. Nonetheless, we would like to point out that we have used common notations surrounding the well-studied subject of curvature, and have also suggested Guggenheim [15], Olver [26], and O'Neill [27], for further reading. These are classical texts on the subject and are widely used in undergraduate and graduate courses. The definitions used are standard and can be found in many textbooks, including the three previously mentioned. This also makes it hard for us to understand what the reviewer means by 'lack of rigor' -- the mathematical section does not contain heuristics or sketches, we have given precise definitions which serve to simply define our tools.
> >
> > To answer the specific questions of the reviewer:
> > - (1) We believed definitions 1 and 2 did not need to be “titled”, as their content is rather explicit and straightforward: def. 1 introduces the shape operator, while def. 2 introduces the principal, Gauss and mean curvatures, which are nothing more than eignenvalues, determinants, and traces.
> > - (2) k1 and k2 are the eigenvalues associated to the orthonormal eigenvectors e1, e2. While it is explicitly mentioned in definition 2, we will specify these terms directly at the end of definition 1 in the revisions, to avoid any confusions.
> > - ⁠(3) DF is a standard notation denoting the differential of a map F. We use the same notation throughout the paper, as so, p is a point in S, and the subscript p refers to the map at point p -- standard notation. However, we are willing explicate this in revisions of our paper.
> > - (4) TS is the standard notation for the tangent bundle of S. We are willing to explicate this in the revisions of our paper. S_hat is introduced as one of the two surfaces referred to on the same line 139. We struggle to see how else it could be introduced.
> > - ⁠(5) Although we fail to see how the indexing of equations is a big problem (given that no equation is referencing another), we will address this in revisions of our work to facilitate reading. Regarding ⁠g_s, it is introduced and defined on line 138, as the Riemannian inner product induced on TS by the euclidean inner product.
> > - (6) We believe F and eta have been introduced in a way that the difference between them is clear: F is an isometry of R^3, while eta is an isometry between surfaces. We already introduce them both in writing and formula -- we struggle to understand what more is needed here.
> > Overall, we believe all objects have been properly introduced. However, we will happily explicate some common notations used in our work to help readers less familiar with them.
> > We hope we answered the reviewer’s questions more specifically in this comment

---

### Official Review · Reviewer_LGgy · 2024-07-10

**Soundness:** 2
**Presentation:** 3
**Contribution:** 2
**Rating:** 6
**Confidence:** 4

**Summary:**

This paper proposes to use surface curvatures as input to neural networks to improve performance on 3D tasks. The main hypothesis is that as the curvatures are intrinsic properties of the surface, they will enable more effective learning for the relevant tasks. The paper carries out some evaluation comparing performance of network with curvatures as input against existing approach that uses point coordinates or other intrinsic properties as input. The proposed approach achieves better results on a few category of tasks.

**Strengths:**

- The paper carries out evaluation with some existing methods to demonstrate the benefit of using surface curvature as input.
- The paper is overall well-written and easy to understand.

**Weaknesses:**

- The use of surface curvature in neural networks is not really new, so the novelty is limited.

- The paper fails to discuss some inherent limitations when using curvatures as input. e.g.
   *) Surface curvature requires 2D manifold structure. In a lot of applications, we are dealing with point clouds rather than a well-defined surface structure such as meshes. On point clouds with thin structures, some points not in the neighborhood of a point on the manifold may be closed actually close to the point in 3D space, causing difficulty in correcting identifying the neighbors and affecting the accuracy of curvature computation.
   *) As a high-order differential property, curvature is sensitive to noises.
   *) The signs of curvatures are dependent on the orientation of the surface (i.e., among the two possible normal directions, which one points to the outside and which one points to the inside). Given a point cloud, it is non-trivial to properly orient the normals to enforce a globally consistent orientation. If some points on the point cloud have incorrect orientation, their curvatures will have negated signs and may mislead the neural network.
   These real-world scenarios can cause difficulties when using curvature rather than point coordinates as input to neural networks. There is insufficient evaluation on such real-world cases.

- In recent years, transformers have shown better performance than convolution networks on many tasks. There is a growing body of transformer-based networks for 3D tasks (see https://github.com/lahoud/3d-vision-transformers), but there is no comparison with such methods in this paper.

**Questions:**

- How well does the method perform when dealing with noisy data?
- How well does the method perform when compared against transformer-based architectures?

**Limitations:**

There is no discussion on limitations. In particular, the dependencies on orientation, the difficulty of orienting the surface globally, and the sensitivity to noises are not discussed sufficiently.

---

> ### Author Rebuttal · Authors · 2024-08-07
>
> We thank the reviewer for their work and time.
>
> We clarify misunderstandings and respond to the questions raised below:
>
> > "The main hypothesis is that as the curvatures are intrinsic properties of the surface, they will enable more effective learning for the relevant tasks."
>
> The idea that curvature should be the representation of choice is not, as stated by the reviewer, solely for its intrinsic properties. Many other methods are intrinsic, such as HKS, which we show is outperformed by curvature.
>
> >"The proposed approach achieves better results on a few category of tasks"
>
> The inflection of this statement is misleading, especially since we achieve better results on *all* categories of tasks competing against the state-of-the-art neural network architectures with state-of-the-art surface representations.
>
> > "The use of surface curvature in neural networks is not really new, so the novelty is limited."
>
> Our methodology and results *are* new and were missed by the rest of the community. What is  important is novelty of results, not novelty of subject matter, since otherwise this point of view shuts down almost all avenues of scientific inquiry. We also make sure to reference papers that use curvature in our work, and explain clearly how we differ from them.
>
> >"In recent years, transformers have shown better performance than convolution networks on many tasks..."
>
> Although they have shown tremendous improvements in the NLP domain, one could argue the verdict on transformers outperforming CNN in image processing is not out yet. More so, regarding shape data, which is the context of our work and results, we have selected three architectures that are the most used in the domain and can undeniably be considered the state-of-the-art in neural networks for shapes.  Nonetheless, we are open to including a comparison with one of the emerging 3D transformer architectures in revisions of our work; as we show that curvature representation outperforms other methods on a variety of architectures, we believe a transformer model would also benefit from such representation.
>
> >"The paper fails to discuss some inherent limitations when using curvature as input... In a lot of applications we are dealing with point clouds rather than a well-defined surface structure such as meshes."
>
> It is true that point clouds are different from meshes. But there are several things to consider here: (1) In our paper, and within the domain of surface processing, we are focusing on shapes for which standard representation is a mesh, as mentioned explicitly in our paper. Application domains making use of mesh shape data are numerous, and large, including medical imaging, biology, architectural artifacts, so the context in which our results are valid is not at all limited. (2) We believe curvature as shape representation could be extended to noisy point cloud data. Perhaps more so than other representations, since discrete curvature knows a long history of research specialized in noisy data. In the paper we reference methods (that have been implemented and tested) from geometric measure theory, that can be used to define curvature for point clouds without an obvious orientation of normals everywhere.

---

> > ### Comment · Reviewer_LGgy · 2024-08-13
> >
> > The rebuttal has addressed my concerns. I am happy to raise my rating to 6.

---

> > > ### Author Response · Authors · 2024-08-14
> > >
> > > We thank the reviewer for taking into consideration our rebuttal, and are happy they raised their rating.

---

> ### Comment · Area_Chair_iouh · 2024-08-11
> **Any comments?**
>
> Dear LGgy. The end of the discussion period is close. I would be grateful if you provide a feedback regarding authors’ answers to your review.

---

### Official Review · Reviewer_qKsN · 2024-07-14

**Soundness:** 3
**Presentation:** 3
**Contribution:** 4
**Rating:** 7
**Confidence:** 3

**Summary:**

The paper proposes to use curvature instead of previous surface descriptors in neural networks that process shapes. They show that the principal curvatures and/or mean curvature are better surface descriptors for three very different neural network pipelines, and is much faster to compute. They point out previously known properties of curvature for why curvature makes for such good shape representation.

**Strengths:**

- While there is clearly performance improvement, it is impressive that the benefit is clear on three very different neural network pipelines
- Furthermore the simplicity and low dimension show of the proposed descriptors really demonstrate their effectiveness
- Reduced computation time scaling means it can be scaled to very large sampling
- "The performance of each representation is strongly dependent on the chosen implementation. We have tried to be as fair as possible by not developing our own implementations of existing work and instead using implementations which have already been tried, tested, and validated in the literature" - Very happy with with mentality, this is very important and a lot of papers in the literature do not do this.

**Weaknesses:**

- The reduction in performance for the principle curvatures for classification is a bit puzzling, especially since Gaussian curvature can trivially be computed from the principal curvatures. Surely the complexity of Delta Net and Diffision Net can cope with that, but there is a large performance reduction compared to HKS. The paper mentions that it indicates that Gaussian curvature interacts better with pooling, but that is not really justified. In the task HKS performs a lot better than shot16/64 indicating that there is some difference in the task. Given that this paper is more of a discovery of the benefit of a well known quantity, I would expect more analysis into this. Especially since for the other two tasks, Gaussian curvature performs worse than the principle curvatures, and intuitively this made sense as it decreased the amount of information, but then discarding this argument for the classification task is not appropriate.
- It would be nice to also benchmark against the pipeline PointNet++ recommended, the "linear combination of HKS, WKS and Gaussian curvature, followed by a PCA projection, leading to a 64 dimensional feature per point"

**Questions:**

-

**Limitations:**

None given, and in the checklist have said it is not necessary given it is a comparative study. However given they also try to motivate why curvature is approriate theoretically and it depends whether to use principle curvature or Gaussian curvature, it would be nice for them to either try to identify when to use principal or Gaussian curvature(s) (theoretically or hypothesis-wise)

---

> ### Author Rebuttal · Authors · 2024-08-07
>
> We thank the reviewer for their positive comments, and for their very valuable remarks.
>
> We acknowledge that our results raise some interesting and important questions that are not answered in the work, as pointed out in the 'weaknesses' section. However, we do not see these as weaknesses of the work: discovery has always preceded explanation in science and it's important that discoveries are shared with the community and circulated. The fact that there are not easily answered questions arising from the discovery is an invitation to investigate our constructions further, which we believe to be a scientific strength.
>
> Nonetheless, we are grateful to the reviewer for their suggestions, that represent excellent directions for future work. We will make sure to include these discussion points in revisions of our work.

---

> > ### Comment · Reviewer_qKsN · 2024-08-10
> > **Keeping my original score, but hope the authors are interested in improving their paper.**
> >
> > I'm not sure I agree with the rebuttal from the authors in regards to my weaknesses. Yes it is important that discoveries are shared with the community and circulated, such as the one presented in the paper, but that does not mean only a surface level investigation of the discovery is needed before circulation, and not being interested to investigate or at least discuss further is not a scientific strength.
> >
> > While I did not expect the authors to necessarily run more experiments based on the weaknesses I listed, I was hoping for at least some discussion or thoughts about them. In regards to the first weakness, do you think that a) the complexity of Delta Net and Diffision Net do not have the ability to compute Gaussian curvature from principal curvatures, or b) they do have the representation ability but they never find Gaussian curvature or some equally useful information derived from principal curvatures?
> >
> > I also had a look at the other reviews. Reviewer LGgy has a point about the difficulty of estimating curvature robustly, and it makes sense that at least a short discussion about this would be beneficial to the paper, and I hope the authors add this. However the experimental results show that at least for the experiments given, current standard methods (like the one from [30] used in the paper) are sufficient for various interesting datasets. I don't think that it is necessary to specifically give results with noisy data, and especially not necessary to give results with transformers.
> >
> > I am keeping my original score of 7, but hope the authors are interested in improving their paper.

---

> > > ### Author Response · Authors · 2024-08-14
> > > **Response and additional comment**
> > >
> > > We thank the reviewer for this additional feedback.
> > > First, we would like to apologize if the reviewer was under the impression that the questions they have raised have not been considered. On the contrary, they have stimulated discussion amongst us and we agree the suggestions will improve the paper. We we will be happy to include a discussion section on the points raised by the reviewer and summarised below:
> > >
> > > Benefits of Gaussian curvature:
> > > - ⁠We believe the initial comments made about better understanding why gaussian curvature could be better than principal curvature are interesting and relevant; it is an important point that will add value to the paper's thesis, and we thank you for highlighting it.
> > > - We could also hypothesize that the size of meshes - rather than the task of classification - could be an important factor, since the SHREC dataset we have used in the paper also happens to be the one that contains the smallest sizes of shapes.
> > > - As we do not have a strong analytical understanding of neural network behaviour, let alone an understanding of their action on shapes, it is hard to give a guideline as to when should we use gaussian against principal curvature or to give a scientific discussion that would not be purely hypothetical/speculative, without any experiments and hypothesis testing as support.
> > > - However, we will be happy to discuss potential experiments that could help tackle this question, e.g. benchmarking both curvatures on different classification datasets with ranging sample size and size of meshes.
> > >
> > > Representation ability of the neural networks:
> > > - ⁠We believe that, at the minimum, delta net and diffusion net are flexible enough to "easily reach" gaussian curvature from principal curvature. In fact, they can probably reach it from the extrinsic coordinates representation: in terms of shape information, each representation is complete in a sense; that is, they contain all the geometric information needed to fully describe the shape. We do not create more information when using curvature, it is just "presented" in a compact, yet information rich, form which the neural network uses to process the shape. In addition, principal curvature and gaussian curvature are indeed very close to each other, but the HKS is as well, it is essentially computed from the shape Laplacian, which is related to curvature.
> > >
> > > Noisy data:
> > > - First, we thank the reviewer for emphasizing both the fact that added experiments on noisy data would be beside the point (since we work with known shapes), and that there is no need for comparing with transformers based architectures.
> > > - As for integrating curvature in a noisy data scenario: curvature has a long history of research, with numerous discretisation methods and implementations, each designed to be robust to different scenarios. We believe that noisy data mostly implies a scenario where the orientation of the shape is missing, and in this case we point out that curvature as defined, and implemented, from geometric measure theory should be the best option, as it doesn't rely on the normals. We have only slightly touched upon this point in the paper but agree that elaborating on it, specifically in section 3.2, would improve our paper.
> > >
> > > We hope we answered the reviewer’s questions more specifically in this comment.

---

### Author Rebuttal · Authors · 2024-08-07

We would like to first thank the reviewers for their time, and thoughtful comments and questions that will certainly help improve the revised paper. In particular we thank the reviewers for acknowledging the qualities of our work:
\begin{equation*}
    \text{"... it is impressive that the benefit is clear on three very different neural network pipelines."}
\end{equation*}
\begin{equation*}
    \text{"... a lot of papers in the literature do not do this."}
\end{equation*}
\begin{equation*}
    \text{"The paper is overall well written and easy to understand"}
\end{equation*}
\begin{equation*}
    \text{"The paper works on a relatively meaningful topic in geometric deep learning..."}
\end{equation*}

We are particularly happy with the comments of reviewer qKsN who perfectly grasped the main message of our work, and highlighted some interesting questions.

Although reviewers LGgy and ifJ1 have pointed out that the use of curvature in surface processing is not new, this is specifically what makes our contribution all the more valuable and all the more publishable. Despite the existence of related work using curvature, our results were missed by the rest of the community, and the questions our work raise are novel and non-trivial.

Finally, we believe some comments made by reviewer ifJ1 are, unfortunately, value judgements and not scientific judgements, and we are concerned about the extent to which they actually engaged with the work.

We respond point by point to each reviewers' remarks in individual comments below.

---

### Decision · Program_Chairs · 2024-09-25

**Decision:**

Accept (poster)

**Comment:**

To characterize shape properties the authors proposed to use the principal curvatures and/or mean curvature. They demonstrated that these descriptors can be efficiently computed and leads to a significant increase in performance on segmentation and classification tasks for three very different neural network pipelines when used as input. The proposed recipe is very easy to implement.

The main achievement of the article is a very simple practical recipe. Since the approach is empirical, its validity should be assessed under different conditions as much as possible. Despite the positive reviews, I would insist on adding additional important experiments, see comments below.

In fact,
- the authors performed testing using relatively old and not very representative datasets such as shrek
- they did no study how robust the results are with respect to changing the hyperparameters of the considered neural network architectures
- the authors did not study to what extent the results change in the presence of noise in the data, how this effect depends on the noise level. It is important to discuss how to deal with noisy data and to which extent the noise corrupts the observed effects
- is this approach suitable only for 3d shapes? will it work efficiently for 3D models of mechanical parts?

I personally would recommend to consider more datasets to verify the proposed approach, e.g. using so-called ABC dataset (collection of one million Computer-Aided Design (CAD) models) to predict sharp features (see the follow up paper DEF of the same authors).